# CCHD Screening Implementation Efforts in Latin American Countries by the Ibero American Society of Neonatology (SIBEN)

**DOI:** 10.3390/ijns6010021

**Published:** 2020-03-14

**Authors:** Augusto Sola, Susana Rodríguez, Alejandro Young, Lourdes Lemus Varela, Ramón Mir Villamayor, Marcelo Cardetti, Jorge Pleitez Navarrete, María Verónica Favareto, Victoria Lima, Hernando Baquero, Lorena Velandia Forero, María Elena Venegas, Carmen Davila, Fernando Dominguez Dieppa, Taína Malena Germosén, Ada Nidia Oviedo Barrantes, Ana Lorena Alvarez Castañeda, Mónica Morgues, Aldana Avila, Diana Fariña, Jose Luis Oliva, Erika Sosa, Sergio Golombek

**Affiliations:** 1Neonatology, Ibero-American Society of Neonatology (SIBEN), Wellington, FL 33414, USA; susana.rodriguez@siben.net (S.R.); alejandro.young@siben.net (A.Y.); lourdes.lemus@siben.net (L.L.V.); ramon.mir@siben.net (R.M.V.); marcelo.cardetti@siben.net (M.C.); jpleitez@gmail.com (J.P.N.); veronicafavareto@hotmail.com (M.V.F.); limamv@hotmail.com (V.L.); hbaquero@uninorte.edu.co (H.B.); lorenavforero@gmail.com (L.V.F.); davilacarmen@hotmail.com (C.D.); fddieppa@infomed.sld.cu (F.D.D.); tainamalenag@yahoo.com (T.M.G.); ada2552@ice.co.cr (A.N.O.B.); alacfe@yahoo.es (A.L.A.C.); monica.morgues@siben.net (M.M.); aldasolav@yahoo.com.ar (A.A.); diana.farina@siben.net (D.F.); joseloliva@hotmail.com (J.L.O.); sergio.golombek@siben.net (S.G.); 2Dean School of Medicine, Universidad del Norte, Km. 5 Vía Puerto Colombia, Barranquilla 081007, Colombia; 3Neonatology, Medicina Alta Complejidad S.A. (MACSA), Hospital Niño Jesús Carrera 75 N° 79B-50, Barranquilla 080001, Colombia; malenavenegas@yahoo.es; 4Pediatric Cardiology, Children’s Hospital, Hermosillo 83100, Mexico; erikaflorei1@gmail.com

**Keywords:** pulse oximetry, congenital heart disease, neonate, hypoxemia, Latin America, Ibero-American Society of Neonatology

## Abstract

Congenital heart disease (CHD) is among the four most common causes of infant mortality in Latin America. Pulse oximetry screening (POS) is useful for early diagnosis and improved outcomes of critical CHD. Here, we describe POS implementation efforts in Latin American countries guided and/or coordinated by the Ibero American Society of Neonatology (SIBEN), as well as the unique challenges that are faced for universal implementation. SIBEN collaborates to improve the neonatal quality of care and outcomes. A few years ago, a Clinical Consensus on POS was finalized. Since then, we have participated in 12 Latin American countries to educate neonatal nurses and neonatologists on POS and to help with its implementation. The findings reveal that despite wide disparities in care that exist between and within countries, and the difficulties and challenges in implementing POS, significant progress has been made. We conclude that universal POS is not easy to implement in Latin America but, when executed, has not only been of significant value for babies with CHD, but also for many with other hypoxemic conditions. The successful and universal implementation of POS in the future is essential for reducing the mortality associated with CHD and other hypoxemic conditions and will ultimately lead to the survival of many more Latin American babies. POS saves newborns’ lives in Latin America.

## 1. Introduction

Latin America is a region of the Americas where languages derived from Latin and Romance languages are primarily spoken; it includes part of North America (Mexico), Central America, the Caribbean, and South America.

Determining the true incidence or prevalence of congenital heart disease (CHD) for Latin America as a whole has been a true challenge. In South America, the frequency of CHD was estimated to be 9‰ live births, according to Pedra et al. [1]. In Latin America, the reported incidence is variable, according to the Latin American Collaborative Study of Congenital Malformations (ECLAMC). For example, in Colombia, a very low prevalence of 1.2‰ live births [2] was recorded between 2001 and 2005. In Guatemala, congenital heart disease was reported as the most common malformation present at birth, with an incidence close to 1% (8 to 11 of every 1000 newborns alive), causing 6% to 8% of infant deaths in children under one year [3]. These and other differences in rates of CHD reported in the literature can be partly explained by inadequate access to medical care and inequalities in diagnosis and therapy, differences in the diagnostic criteria, difficulty in obtaining an accurate diagnosis of all children with the disease (which is virtually impossible owing to unknown prenatal deaths), and insufficient data bases and collaboration among nations and true regional variations. When looking at all of the studies available in the literature, an incidence of 9–11‰ live births appears to be a fair approximation of the true incidence of CHD in Latin America [4]. Therefore, the existing inaccuracy of indicators does not permit us to know the exact incidence of CHD in Latin America. Notwithstanding, CHD represents the third most common cause of neonatal mortality and it is estimated that approximately one in every 40 deaths in children under one year is due to CHD. In addition, without some type of medical intervention, at least 14% of children born with CHD do not survive the first month of life and 30% die during the first year, which affects pediatric morbidity and mortality globally. Furthermore, in Argentina, it was estimated that more than 7000 children are born every year with some type of CHD. About 30% die without being diagnosed, mostly before the first month of age [5]. In Mexico, 3593 deaths due to CHD were reported in one year in children less than 12 months of age; more than 50% of those deaths occurred in the neonatal period [6].

Neonatal oxygenation and hypoxemia are not easy to assess in clinical practice. Most of the available evidence derived from randomized clinical trials, physiology, and large descriptive trials has formed the foundation for oxygen saturation monitoring in the neonatal period. Based on the landmark articles by de-Wahl Granelli [7,8,9], Ewer [10,11,12,13,14], Martin and Hom [15,16], Sola [17,18,19,20], and others, it has become clear that infant safety and outcomes can be improved by assessing oxygenation with adequate saturation monitoring. Back in 2016, we published the results of the “IX SIBEN Consensus on early detection of diseases that course with neonatal hypoxemia using pulse oximetry” [21], which is a document developed by neonatologists and nurses from Latin America, with the expert opinion of Dr. Andrew Ewer. We formulated recommendations to implement programs with pulse oximetry for the early detection of pathologies and/or diseases that have neonatal hypoxemia in asymptomatic and apparently healthy newborns. SIBEN recommended the implementation of early detection programs with appropriate pulse oximeters in every neonatal center in Latin America.

General knowledge on the need to carry out the early detection of critical CHD (CCHD) exists amongst physicians, though there is no proper training about pathophysiological aspects that can be assessed with pulse oximetry during the first 24 h of life. Pulse oximetry screening (POS) is a simple, noninvasive, and relatively very low-cost method for the early detection of serious neonatal illnesses. 

Our objective here is to describe early POS CCHD screening implementation efforts in Latin American countries guided and/or coordinated by SIBEN, as well as the unique challenges and infrastructure in these countries for providing amplification of implementation, which we consider may be of great benefit to newborn infants, the international screening world, and various countries that are working to overcome implementation challenges. All regions in this manuscript are at sea level or at less than 250 meters above sea level, so we cannot comment on early POS at altitude.

## 2. The Ibero-American Society of Neonatology (SIBEN)

The Ibero-American Society of Neonatology (SIBEN) is a non-profit public charity organization with a clear mission registered in the USA over 15 years ago [22]. Its main mission is to collaborate to improve the neonatal care of ill newborns and their outcomes in Latin America through the education of nurses, MDs, and other healthcare workers involved in their care. The development of SIBEN’s Clinical Consensus is among the many actions performed in the Latin American region to advance activities with the dynamic and collaborative participation of neonatal nurses, MDs, and other healthcare workers. The process was described 11 years ago with the publication of the first SIBEN Clinical Consensus [23]. To date, SIBEN has completed 12 consensuses on different clinical topics. The ninth Clinical Consensus was on POS [21], including the participation of 45 neonatologists and neonatal RNs from 16 countries in the region who worked collaboratively during one year via the internet. Subsequently, the consensus group met in SIBEN’s annual congress in Asunción, Paraguay, where the consensus statement on POS was developed and finalized. For the present manuscript, to provide information on the current situation of POS in Latin American countries, 19 neonatal health care professionals of 12 countries collaborated to provide information.

## 3. CCHD Screening Implementation Efforts in Latin American Countries

The following information is presented in alphabetical order for the following 13 countries in Latin America: Argentina, Bolivia, Chile, Colombia, Costa Rica, Cuba, Dominican Republic, El Salvador, Guatemala, Honduras, Mexico, Paraguay, and Peru. In a few of them, whenever available, regional information is also provided.

### 3.1. Argentina

Argentina is a country in South America with a population of over 44,600,000. Its capital is Buenos Aires, and the greater Buenos Aires area has over 13,000,000 inhabitants; being the largest urban area in the country, the second in Latin America, and one of the 20 largest cities in the world.

We will first present an overall view on POS in Argentina and then report the findings obtained in two cities.

In 2018, there were 685,394 live births; 59.6% of these births occurred in health institutions belonging to the public sector and 40.4% in the private sector. The neonatal mortality rate for that same year was 6‰, with extreme values ranging from 4.1 to 10.3‰, according to different jurisdictions, with persisting inequities. One in four causes of neonatal death are secondary to congenital malformations, and CHD is the first cause of death among them. In Argentina, it was estimated that 55% of all neonatal deaths are due to reducible causes [24].

Approximately 7000 children are born every year with CHD. About 50% of these children will require surgery in the first year of life and two thirds of them will survive with a timely diagnosis and treatment. Some local studies analyzed the surgical complications of this population, but they did not include those related to a timely or early diagnosis, transfer, and preoperative stabilization [4].

Given the contribution of this disease to neonatal and infant mortality, a National Congenital Heart Disease Program was created to coordinate the referral, transfer, treatment, and follow-up of children with CHD without insurance coverage. However, this program is for children with a confirmed diagnosis of cardiac malformations, and does not contemplate POS.

The Clinical Consensus of SIBEN on POS for neonatal hypoxemic conditions [21] has been very useful and shows that a fetal ultrasound, even in good hands, has a low sensitivity. Furthermore, it is costly and is lacking or not always available in many regions of Argentina. In addition, a neonatal clinical examination does not detect 20% or more of critical CHD and the observation of cyanosis is not effective in detecting hypoxemia. The Clinical Consensus of SIBEN [20,21] and various other publications (some quoted in this manuscript) clearly show that POS is a simple, low cost, and effective intervention for the early detection of serious neonatal hypoxemic diseases, in addition to CCHD.

There have been efforts to implement the early detection of CCHD, such as government regulations and diffusion of the algorithm, for performing pre- and postductal POS in all maternity hospitals in the country. We have no doubt that SIBEN’s collaboration with education in various regions and its consensus has been very helpful. In infants in the public sector, when the test is positive and CCHD has been confirmed by a Pediatric Cardiologist, communication with the national CHD program is established. In addition, legislation has been passed, like Law 25,929 of Humanized Childbirth (also called Respected Childbirth), specifying that it is mandatory for all newborns to be hospitalized for at least 36 h after birth. Moreover, there are currently projects working on incorporating CCHD into the mandatory list that exists in the Neonatal Screening national law # 26,279, which already includes screening for the detection and subsequent treatment of phenylketonuria, neonatal hypothyroidism, cystic fibrosis, galactosemia, congenital adrenal hyperplasia, biotinidase deficiency, retinopathy of prematurity, Chagas, and syphilis, but POS is not considered. There was a draft amendment to article 1 to include POS, but at the end of 2019, it lost parliamentary status. The Ministry of Health has made some recommendations in this regard. It is also difficult to obtain accurate data. All of this makes POS implementation variable.

In 2015, the Secretary of Maternal Infant Health conducted a survey gathering information from various authorities in 66 maternity hospitals of the public sector in five provinces of Argentina (San Juan, Jujuy, Chaco, San Luis, and Tucumán). It was found that the implementation of POS was irregular, generally infrequent, and lacked systematized protocols, due to difficulties in the provision of saturometers and lack of human resources assigned to the task. There was significant variability in the responses, but in general, POS was not performed systematically and was contingent on the doctor on call or the nurse on duty. Nowadays, in many public and private institutions, POS is still not done systematically, but instead depends on the health care team that is providing care to healthy newborns on that particular day. In Argentina, the two greatest difficulties are the lack of suitable pulse oximetry monitors and sensors, and a deficit of personnel assigned to perform POS universally. In terms of the private sector in the city of Buenos Aires, we have information from three NICUs where all newborns have POS early, before discharge. To briefly mention one of them, with close to 2500 live births per year, POS is performed for all live births, as it should be. In this past year, one baby with CCHD was found. The baby was referred for surgery and did well. Other infants with CCHD are diagnosed by a fetal ultrasound and born elsewhere. Every year, their maternity hospital identifies 3–5 babies early with hypoxemic conditions other than CCHD. In these private sector maternity hospitals in Buenos Aires, there is no charge to the families for POS, which is implemented as part of routine newborn care.

We believe that there is an urgent need for the further expansion of POS in the nation. To date, there are no studies or data available to know with certainty the status of the implementation of POS before neonatal discharge, but, as is true of other indicators of perinatal health, the impression is that the variability in POS is wide, with some centers performing a well-established program, some having significant inefficiencies, and some not conducting POS at all. As a representation of this, POS implementation is in general low and intermittent, according to different days of the week and with less compliance in the maternity hospitals of the public sector than in the private ones.

The inter-center disparity in neonatal outcomes is dreadful in the country. For example, looking at the rates of early neonatal mortality, the center with the highest mortality rate was 16 times higher than the center with the lowest rate. The failures were mostly related to deficits in the availability and training of human resources and coordination of care [25].

In the particular case of the early detection of heart disease or hypoxemic diseases by pulse oximetry, the lack of nursing resources or low availability of nursing h in rooming-in areas of maternity hospitals and limited availability of equipment (pulse oximeters) seem to be the main barriers to implementing this beneficial clinical practice. In many places, in the face of scarce resources, the NICU is prioritized for the use of pulse oximeters and/or the assignment of nurses and, therefore, POS “suffers” in normal newborn nurseries.

One of the questions, for which no easy answer exists, would be related to why other more complex and more expensive (and less effective) practices are incorporated more easily. In this worrying reality, where interventions of proven effectiveness are not adopted to be applied in practice, research on policy implementation [26] could help understand the context, evaluate the performance, guide implementation, and facilitate the strengthening of health care systems.

Nevertheless, the good news is that the quality of care continues to improve, despite all of the difficulties, and some centers have implemented POS, as shown below, in two cities of Argentina.

#### 3.1.1. City of San Luis

With almost 180,000 inhabitants, the city of San Luis in Argentina has two NICUs and high acuity maternity services, including one from the private sector and one from the public sector. The number of births per year is about 4000.

One year ago, with the collaboration of SIBEN, both hospitals started POS implementation. There were some challenges. In particular, one was the nursing staff’s lack of information on the subject. Therefore, training began using a step-by-step approach to follow the program as described in SIBEN’s Clinical Consensus [21]. This took approximately 2–3 months before moving on to the first clinical steps for the early detection of hypoxemic diseases with POS.

It was possible to acquire a Masimo SET^®^ monitor with special software for the detection of hypoxemic diseases; the complication is the high cost of the non-reusable sensors. In addition, the early discharge of both vaginal and caesarean section births also adds to the challenge.

It has now been 6 months since the full implementation of POS has been systematically achieved for the early detection of hypoxemic conditions, not necessarily just critical CHD.

POS is now routine and universal in San Luis province, with a very good reception by families. The main difficulty in conducting POS before discharge has been the early discharge that occurs for some newborns. The private NICU at Clínica y Maternidad CERHU has been able to work this out fairly well in the rooming-in area. Over 1400 infants have been screened so far. Four hypoxemic infants have been detected in this short period of time; none had CCHD, but all of them needed supplemental oxygen and fortunately had good outcomes.

A team of MDs and RNs from SIBEN visited San Luis, in November 2019, to provide a 3-day educational workshop for neonatal nurses and neonatologists. Everyone is now fully aware of the importance of screening and they seem very excited about this program as they see its value for improving the quality of newborn care. Furthermore, in discussions and conversations with the two directors of neonatal services and with the Minister of Health, a full commitment to the continuity of POS has been secured.

#### 3.1.2. City of Rosario

The city of Rosario is a major city in the province of Santa Fe, Argentina, and is the third most populous city in Argentina. The greater metropolitan Rosario has close to 2,000,000 inhabitants, with over 22,000 live births per year. Deliveries and newborn care are provided in both private and public hospitals. 

By the end of 2016 and early 2017, the Provincial Hospital of Rosario started activities aimed at implementing POS. At the beginning, there were several challenges, difficulties, and obstacles. The main drawback was the human factor, mainly skepticism and laziness, with struggles in assigning personnel and establishing a schedule for performing POS, especially during weekends and holidays.

SIBEN, through efforts of the Hospital Provincial de Rosario, started to collaborate with neonatal education in the city. Teams of MDs and RNs from SIBEN provided educational workshops and a particular educational program, which SIBEN calls “Dialogues in Neonatal Medicine”, in 2017 and 2018. In the last few years, we developed a Clinical Consensus on POS [21] and trained health care personnel for its implementation. By early to mid 2018, neonatal nurses were assigned to perform CCHD screening. One hurdle was to ensure that the equipment was available all the time solely for POS. It took until early 2019 to obtain a Masimo SET^®^ pulse oximeter and its sensors exclusively for such a purpose. Now, it has also incorporated the software designed specifically for POS (called “Eve”). Currently, all apparently healthy newborn infants are examined by neonatologists in the rooming-in area, where they are kept with their mothers, and POS is performed by the assigned RN there. The hospital has a section of pediatric cardiology with available echocardiography and follows the SIBEN’s IX Consensus for the early detection of diseases with neonatal hypoxemia using pulse oximetry.

Due to the mentioned difficulties, initially, POS was only intermittent. No CCHD was diagnosed during that period. Of the infants screened, 28 newborns failed the first test, 25 of them passed the repeat test, and 3 newborns needed the test to be repeated twice. One infant had a positive POS and was admitted to the NICU; the echocardiogram was normal, but the infant required supplemental oxygen for 5 days due to severe transient tachypnea. In addition, seven babies with normal POS had an echocardiogram due to a heart murmur: four of them had a small muscular ventricular septal defect and three were normal. During the same time period, the NICU received three newborns referred from other institutions with complex CCHD in a critical status; none of them had had POS. One of these infants was referred at 6 days of age, was found to have aortic coarctation, and died within a few h of admission. This led to the initiation of steps towards the implementation at other institutions in Rosario.

During 2018, POS started to function better at the Provincial Hospital, but some difficulties still persisted, mainly related to personnel issues. There was still some resistance, no acceptance of the added clinical value of POS, and a lack of support for the program by some MDs and RNs. Sometimes, absence of commitment and responsibility are still present. This led to interference with universal screening and/or to incomplete data recording in screened infants. In 2019, all infants were screened every day of the year. Finally, technical issues have also been detected on occasion, like freezing of the monitor and imperfect adhesion of the sensors. 

Since the middle of 2019, POS has been performed for all healthy newborns in all public hospitals of Rosario, but only in two of the private institutions, where they charge the medical insurance or social security for POS. When the charges are not covered, if the parents refuse to pay for the screen, POS is not done. Three other private neonatal departments do not perform POS, but refer all healthy newborn infants to a consult with pediatric cardiology, where an evaluation is done to rule out CHD, sometimes including a “screening” echocardiogram. This is at a cost which, if not covered by private medical insurance or social security, the parents have to pay. SIBEN, of course, does not support these practices. 

It is hoped that as of 2020, accurate universal screening will finally be accomplished in all hospitals, mostly in private ones, and that data will be available for all apparently healthy newborn infants.

### 3.2. Bolivia

Bolivia’s current population is 11,633,371 inhabitants. It is estimated that 243,000 children will be born in 2020 (UNICEF data).

The constitutional capital of Bolivia is the city of Sucre, which is the seat of the judicial body, with 300,000 inhabitants. The city of La Paz is the seat of the executive, legislative, and electoral organs and its population is 2,900,000. Santa Cruz de la Sierra, with a population of about 2,100,000, has 76,000 births (National Statistics Institute from Bolivia), or a third of all births, in Bolivia. Dr. Percy Boland Women’s Hospital, a large maternity hospital, is located in this city.

Here, we present a summary of major issues and difficulties that exist in the nation and in the major Bolivian cities in relation to POS and CHD.

The total number of births at Dr. Percy Boland Women’s Hospital was 6224 in 2018; 1728 were admitted to the NICU. In reference to hypoxemic conditions, 572 (33.1%) were diagnosed with respiratory distress syndrome, 20 (1.15%) with persistent pulmonary hypertension (PPHN), and 34 (1.96%) with CHD, but underreporting is likely.

Bolivia does not have any program designed by the Ministry of Health for the detection and treatment of CCHD or POS. Existing programs depend on international collaboration, and are focused on treatment and not on the detection of congenital pathologies. The causes for this deficit are numerous, including the absence of a reference surgical center, the lack of specialists, and the very high costs of the implementation of comprehensive programs that can solve complex problems. However, some centers have worked out algorithms for the implementation of the early detection of CHD, but the lack of personnel and equipment, together with the non-resolution of the problems detected, have caused individual and isolated attempts to fail.

At the Percy Boland maternity hospital, a program was designed based on the recommendations of SIBEN, but implementation was not possible for reasons such as insufficient numbers of doctors and pulse oximeters in rooming-in areas, weekends without medical personnel in that area, and high rates of very early discharge due to supersaturation in the maternity area. A pediatric cardiology service is available, but there is no regional cardio-surgical center.

The Women’s Hospital of La Paz has a screening program that has included training, but due to a lack of staff and equipment, it cannot be established on a daily basis, despite the fact that in the Children’s Hospital of the city, there is a cardio-surgical center for the resolution of these pathologies.

In Sucre, about 2 years ago, SIBEN presented a detailed educational program for POS, but they still do not have the staff and/or equipment to do this routinely. They also face the problem of early discharges (before 12 h of age), and the lack of a pediatric cardiologist.

In the city of Cochabamba, at the Public Hospital, there is no such program.

The city of Tarija has a POS program and the necessary equipment. This program involves senior medical students carrying out the screening, but, unfortunately, for babies who have a positive POS test, the lack of a pediatric cardiologist makes it very difficult to conduct timely assessment and treatment.

### 3.3. Chile

In Chile, there are approximately 18,700,00 inhabitants and 250,000 babies are born every year. Of these, 6% to 7% are premature <37 weeks and 1.5% are <32 weeks. About 70% of births occur in Santiago and in the most urbanized areas of the center of the country. CHD affects 9‰ live births and 25% of them need surgical resolution during the first year of life, and usually in the first weeks of life.

The Chilean health reform implemented since 2003, as of 2005, has developed a clinical guide for the early diagnosis of and specific actions for CCHD. Since that date, efforts have focused on prenatal screening and diagnosis, with an emphasis on the neonatal period for the most complex cases. A care network was implemented, centralizing the surgical resolution in the centers with the best results in the country. There were significant efforts involving prenatal diagnosis with the participation of pediatric cardiologists in cooperation with obstetric sonographers.

The 25 to 30 weeks’ gestation interval was selected to perform a bi-dimensional echocardiography with a color doppler when obstetric screening raised the suspicion of heart disease or if there were associated risk factors. There is a good correlation and 5% false positives. Once the case is diagnosed, the patient is treated with the goal of referral for delivery in centers of a greater complexity within the network and/or terminating the pregnancy. If it is suspected that the newborn should be intercepted at birth, the case is referred to the centers designated for neonatal management and surgical resolution.

As there is a significant percentage of prenatal screening failure, the use of pulse oximetry among newborns is recommended to improve the timely diagnosis of CCHD. In the national standard of comprehensive neonatal care published in 2017, it is already recommended, as a norm, to perform POS between 24 and 48 h of life. The implementation of this recommendation has been gradually extended in recent years throughout the country, depending on the resources of each neonatal center. SIBEN’s recommendations were evaluated and reinforced the use of adequate oximetry in the national recommendation for timely and early screening that has been in the process of being implemented in the past two years.

The main difficulties have been the lack of equipment and supplies and the shortage of available h of trained professionals (nurses) for implementing adequate POS in a standardized manner prior to the discharge of all healthy-appearing newborns. This is still a work in progress.

### 3.4. Colombia

Colombia is a country with 48,300,000 inhabitants and about 650,000 live births per year. Barranquilla, where the first screening program in the country for congenital heart disease and other serious hypoxemic diseases in the neonatal period started, is the capital district of Atlántico Department in Colombia. As of 2018, it had a population of about 1,300,000, making it Colombia’s fourth-most populous city.

In the last two decades, meaningful efforts have been made to reduce infant mortality in the country. The implementation of programs, including those specific to timely and quality care during the perinatal period, allowed a decrease in the infant mortality rate from 19‰ live births in 2000 to 13.3‰ live births during the first period of 2019. Government efforts have been complemented by individual interventions originating in scientific societies and the professional practice of neonatology. An example of this is what happened in Barranquilla with the neonatal group at Hospital Niño Jesús, MACSA (Medicina Alta Complejidad S.A.), which, with the support of SIBEN and Masimo Corporation, started to follow the pulse oximetry screening (POS) guidelines described in the SIBEN’s Clinical Consensus [21] to implement the screening program for all births in this hospital, which has about 2400 annual births.

Resource limitations were significant challenges to overcome and it took about two years for the full implementation of the screening program on January 1, 2016. The data of the program, until the end of December 31, 2019, are as shown in Table 1.

Of the 20 true positive cases, 12 infants had various types of congenital heart disease (see Table 1) and 8 had other hypoxemic conditions, including persistent pulmonary hypertension of a variable severity. This represents 40% of the true positive tests and 21% of all positive tests. Therefore, the usefulness of POS is not only for the early detection of congenital heart disease, but also for the early detection of other severe neonatal hypoxemic conditions, which can be treated much more effectively by recognizing them before severe decompensation occurs, as was described extensively in SIBEN’s consensus.

Another challenge that had to be overcome initially was the high number of true false positive results. Most of the 18 true false positive tests shown in the table were observed during the first weeks after the implementation of the screening program, until further education and quality improvement measures produced a sustained reduction of errors. The current true false positive rate is low, as is described in various international publications.

Two newborns died, both with a single ventricle, within the protocol of dignified death and compassionate care. The two infants with total anomalous venous return were sent by air ambulance to another city. In the Caribbean region of Colombia, there are currently no institutions with formally established programs for the diagnosis, treatment, and rehabilitation of congenital neonatal heart disease.

As a secondary but substantial gain from the implementation of CCHD screening, the need arised to acquire an echocardiograph and train two neonatologists in the performance of echocardiograms to confirm or rule out the presence of a structural or functional anomaly in infants with a positive screening test. This training is provided for currently valuable diagnostic and follow-up tools for all critically ill infants. In addition, a support network was generated in Bogota, the country’s capital (led by a pediatric echocardiographer, Dr. Edgardo Vanegas), that can provide an almost real-time opinion via telemedicine in the case of neonatologists’ doubts on echocardiographic findings. 

One of the most recently implemented perinatal health policies in Colombia is Resolution 3280 of 2018. This resolution makes POS for complex congenital heart disease mandatory in the immediate neonatal period. This intervention is part of the cost-effective strategies contained in the comprehensive health care plans for the Colombian perinatal maternal population.

The first cost-effectiveness studies on POS in Colombia and other articles were published in the last few years [2,27,28,29]. It has been reported that although early detection is cost effective, treatment and follow-up are still very expensive for the economic reality of Colombia’s health system. It is our understanding that in Colombia, there are only six centers performing POS: the one described here in Barranquilla, Bolivarian Clinic in Medellin, Clínica del Country in Bogotá, Cesar Clinic in Valledupar, Santa Cruz de Bocagrande Clinic in Cartagena, and Medilaser Clinic in Florencia.

### 3.5. Costa Rica

Costa Rica has a population of just over 5,000,000. Its capital is San José, with a number of inhabitants of 345,000 and in that province, the number is close to 1,400,000. The total number of births per year in the country has been decreasing and is currently around 65,000 [30].

In Costa Rica, POS was initiated in 2014 as a pilot plan in two hospitals (Women’s Hospital and Calderón Guardia Hospital). Initially, it was partially done, since on Sundays and holidays, it was not carried out due to a lack of personnel. In this pilot group, of a total of 899 screened newborns, four had CHD. In 2015, POS was implemented at the San Juan de Dios Hospital at 12 h of age.

With the results obtained from these hospitals, and with the collaboration of SIBEN’s education and its Clinical Consensus [21], POS was established in all public and private health centers in the country. As of August 2016, it was established that POS should be performed between 12 and 24 h of age for all healthy newborns, and it was requested that no child should leave the hospital without having POS. With a total of 33,804 births, 16 infants with CCHD were detected early. A similar number was found to have other hypoxemic conditions early, mainly PPHN and sepsis.

The biggest challenges have been related to the decision of who should perform POS, since most centers have staff problems. In general hospitals of the country, respiratory therapists perform it in the mother’s bed in the rooming-in areas. In smaller hospitals, the nursing staff performs this. The equipment to be used has been unified for the country and Masimo monitors (Irvine, Ca) are used in all the maternity units of the Costa Rican Social Security Fund (28 in total).

### 3.6. Cuba

Cuba is a country with close to 11,500,000 inhabitants and approximately 117,000 births per year. The capital is Havana, with a population of 2,300,000. This is where Gonzalez Coro Maternal-Infant Hospital is—a landmark for improvement of newborn care in Cuba.

Cuba’s infant mortality rate is 4.0‰ live births, the lowest in Latin America, and lower than in many developed or industrialized nations. The analysis of infant deaths reveals that congenital malformations and chromosome abnormalities account for 20% of all infant deaths, with a rate of 0.8‰ live births. CHD represents 65% of all congenital malformations and has a mortality rate of 0.5‰ live births [31]. 

In Cuba, abortion has been legalized for many years and, since a fetal echocardiogram is performed in over 99% of pregnant women, pregnancy is terminated in many pregnancies when the fetus is diagnosed with serious CHD. Additionally, a complete autopsy is performed in every newborn and older child who die in Cuba so, if the diagnosis is missed clinically, it is confirmed in the post-mortem exam. 

It is well-known that POS performed in healthy-appearing neonates, with monitors that can measure with a high specificity and sensitivity through low perfusion and motion, is very valuable for detecting early CCHD [21]. Other hypoxemic conditions that are frequent causes of neonatal mortality or long-term sequelae, like PPHN, pneumonia, sepsis, and other conditions, can also be detected early by POS. POS is more useful if performed early, but always taking into consideration a prenatal ultrasound and detailed physical exam of the newborn. 

In Cuba, SIBEN has been collaborating for many years with education for continuous improvement of the quality of neonatal care, performing two, and up to four, visits per year. In the last few years, the subject of the early detection of neonatal hypoxemia in different territories and provinces of the country has been included in the educational programs. SIBEN has also donated five SpO_2_ monitors and sensors suitable for this purpose. However, to date, there are no fully established programs for POS, since Cuba is not allowed to acquire the necessary monitors and sensors from the United States, due to the blockade that has existed for over 60 years ago, which has recently intensified.

For the early detection of CCHD, Cuba relies on two evaluations: a) a fetal ultrasound, which is performed in almost 100% of pregnant women at no cost, regardless of the place of residence of the mother, and b) a thorough clinical examination. This is possible as 99.9 % of infants are born in institutions where there are trained pediatricians and/or neonatologists who perform detailed exams in apparently healthy neonates during their hospital stay, which is usually 48 h or more. Exams are then performed in an ambulatory setting in all infants in 3 to 5 and 7 days. The national pediatric cardiac network aims to provide the referral and timely management of newborns and infants who are diagnosed with CCHD and CHD.

A few neonatal centers have intermittently performed POS with SpO_2_ monitors donated by SIBEN, since the limited number of sensors available are frequently used to care for critically ill neonates in the NICU. On the other hand, there are absolutely no difficulties or challenges with human resources and infrastructure in the country’s maternal and infant centers. They therefore do not hinder the implementation of SIBEN’s recommendations for neonatal intensive care or POS. The latter, however, will only be able to be universally implemented when the necessary adequate SpO_2_ monitors and sensors, with the specificity and sensitivity described in the published literature, become available. Until then, for CCHD diagnosis, most newborns in Cuba rely on findings of universal prenatal ultrasounds and physical exams.

### 3.7. El Salvador

El Salvador is a country in Central America with about 6,400,000 inhabitants and about 91,000 live births per year, delivered in the hospitals of four systems of health care in the country, which are the Ministry of Health, Solidarity Fund for Health (FOSALUD), Salvadoran Social Security Institute (ISSS), and Military Health Command (COSAM).

The Hospital Nacional de la Mujer (National Women’s Hospital) is the main public level 3 center for perinatal care and is located in San Salvador, which is the capital city. The number of births in 2019 was 7481 live births. SIBEN has had a significant interaction with this hospital in order to collaborate on neonatal quality improvement, and has recently been there three times to discuss SIBEN’s protocol for POS. In this country, POS has not started yet. The National Women’s Hospital will be the first hospital to start POS screening.

The National Women’s Hospital and the Integral Attention Unit for Women, Children and Adolescents of the Ministry of Health are making advances to strengthen the steps for the implementation of neonatal POS for all infants who are born at the 28 maternity hospitals in the country.

There are still significant challenges to overcome, mainly related to resource limitations. Among them are the following:Seven months with a new government. SIBEN has already begun to adopt approaches with the authorities. They seem very open to the implementation of POS (in the previous government, there was not much interest);Resistance, lack of interest, or no positive attitudinal behavior for accepting and making changes (this was also noted with several of the practice changes presented by SIBEN’s clinical consensus on various topics);Extensive need for education and educational materials;Process and method for recording results;Individual hospital budgets for the implementation and ongoing expenses of the POS program;Insufficient funding for necessary equipment;Technical-financial support for implementation.

In 2017, a review of 203 children with cardiac anomalies was carried out. There were 145 major anomalies and 58 minor anomalies. Of the 25 people with significant hypoxemia, the diagnoses were: Truncus arteriosus [7], tetralogy of Fallot [6], pulmonary atresia [4], transposition of the great vessels [4], hypoplastic left heart syndrome [2], anomalous pulmonary venous return [1], and tricuspid atresia [1].

The authorities of the hospital and the Minister of Health seem committed to implementing POS in order to

Detect congenital heart abnormalities early, to influence, in a positive way, medical or surgical treatment as appropriate, in a timely manner, and influence the reduction of neonatal mortality and morbidity;Provide comprehensive and quality care for Salvadoran neonates;Improve the quality of life of children with congenital heart disease in El Salvador.

It is hoped that during the visit of SIBEN’s Medical Director at the end of March 2020, many aspects will be finalized, and that POS for Salvadoran newborn infants will start by the second half of 2020. Additionally, at the time of submitting this manuscript, we found out that, through the efforts and collaboration of SIBEN, the Social Security Institute of the whole country, with over 2600 deliveries per year, plans to start universal POS by the last trimester of 2020.

### 3.8. Guatemala

Guatemala is a country with approximately 18,000,000 inhabitants. It has an annual growth rate of 2.07%. In 2017, there were 381,664 births, increasing in 2019 to 486,497. The infant mortality rate was 25‰ live births in 2019, and the neonatal mortality rate was 17‰.

Health services are provided by three sectors: the Ministry of Public Health, the Guatemalan Institute of Social Security, and the private sector. There are hospitals in all departments of the country, with 22 departments in total, which fail to cover the entire population, due to the distance between the population and health institutions. The national hospital network does not have pediatrician coverage 24 h a day in all hospitals, and the availability of neonatologists is even lower.

In 2011, a thesis paper presented at Universidad de San Carlos de Guatemala (“Late diagnosis of congenital heart disease”, García Vargas and Livengood Ordoñez) was performed at the Pediatrics Department of the Cardiovascular Surgical Unit of Guatemala (UNICAR) with data from 2005–2010. The thesis reported that 55% of CHDs were diagnosed very late, after one year of age. The most common diagnoses were as follows: double outlet right ventricle, tetralogy of Fallot, anomalous pulmonary venous connection, transposition of great arteries, ventricular septal defect, and complete atrioventricular canal defect. It was also demonstrated that the late diagnosis of CCHDs was an avoidable risk of mortality and morbidity.

From 2012 to date, some isolated research studies have been carried out to be able to implement protocols for the early detection of CCHD, including POS. They have been carried out by UNICAR in Guatemala City, in the Metropolitan Hospitals with the highest number of births, and in the interior of the country in two hospitals of the public health system.

However, despite the evidence showing the usefulness of POS for the early diagnosis of CCHD, currently, only one public health hospital and one private hospital in the capital city have established a cardiac screening program, following SIBEN’s Clinical Consensus.

Among the main difficulties in the implementation of POS is the low interest of the health system and of the professionals who care for newborns. This leads to a lack of dedicated personnel, adequate equipment (oximeters and sensors), protocols for the management of these patients, and algorithms for adequate clinical management.

In the whole country, there is only one national public referral center for the treatment of these infants (UNICAR), and there are a handful of hospitals at the private level that have this service. In the interior areas of the country, only two departments have pediatric cardiologists, which is a challenge when detecting a patient with a suspected diagnosis of CHD. Many infants are unable to be transferred or referred for consultation. The socio-economic status of many families often does not allow for them to travel. Although the consultations and surgical proceedings in the public system are at no cost to the families, in many cases, the patients do not make their follow-up appointments due to economic and distance reasons. Additionally, the few specialized centers for treatment are oversaturated, leading to delays, sometimes of months. Therefore, infants either do not get there or arrive with complications that increase morbidity and mortality. The outcomes of infants who are lost are ignored, but, undoubtedly, some die.

Finally, there are not enough adequate pulse oximeters for proper POS and there is no data available regarding other hypoxemic conditions diagnosed early by POS.

### 3.9. Honduras

Honduras is a Central American nation with 9,500,000 inhabitants that reports 140,000 to 150,000 births annually, with a birth rate of 22.8‰ live births and a neonatal mortality rate of 19‰. Tegucigalpa is the nation’s capital, with a population close to 2,000,000.

The four main causes of mortality are prematurity, sepsis, asphyxia, and congenital malformations, including those of a cardiac origin. The estimated incidence of congenital heart disease in Honduras is 9‰.

In Honduras, POS has not yet been approved as mandatory for all newborns. However, due to efforts by specialists in the area and after educational events by SIBEN in the country that attracted the attention of the majority of those who provide neonatal care in Honduras, the discussion has begun to make POS a law at the country level. In addition to this collaboration, SIBEN has donated SpO_2_ monitors to initiate this program in the largest hospital in Honduras—”Hospital Escuela” in Tegucigalpa—where approximately 14,000 to 16,000 babies are born per year. This hospital is a national referral center and 36% of the neonatal mortality in Honduras occurs there. Therefore, it made much sense to start working on POS at this institution.

The first POS experience reported in Honduras was from Hospital Escuela, where there are about 45 to 60 births per day and a physical space that can house around 70 mothers with their babies, so the demand for care requires a rapid turn-around. Following the donation of SpO_2_ monitors by SIBEN, screening was initially started irregularly in 2016 at Hospital Escuela. Subsequently, a review of some data was presented to the authorities to justify hiring more staff and making POS a required test for all healthy newborns prior to the discharge. Of 1221 newborns, seven with CCHD were detected by POS. Five of the seven, or 71%, were boys. After staff training and education, and with the hiring of more staff, POS has now been implemented since 2017.

Following SIBEN’s initiative at this hospital, many other centers in the country started POS informally through the staff’s own efforts. This has not been done continuously and there are no reliable reports of their findings. However, it has impacted the age at which surgical heart disease corrections are made in the country.

In Honduras, virtually all congenital cardiovascular surgery is performed at the Maria Hospital in Tegucigalpa. The Maria Hospital has been reporting a decrease in the average age at which heart patients receive interventions. In 2017, 13% of cases were newborn infants; in 2018, this increased to 18% and in 2019, this increased to 24%. This is even more significant taking into consideration that for three months in 2019, there was no available surgery, since the surgical team travelled to receive further neonatal cardiovascular surgery training abroad.

We found POS to be very useful for the early detection of hypoxemic conditions other than CCHD. In recent months, for example, two healthy looking infants were identified early as hypoxemic and were found to have neonatal sepsis. Others with mild-moderate transient tachypnea and/or pulmonary hypertension were also identified early.

### 3.10. Mexico

Mexico is the most populous Spanish-speaking country, and the second most populous in Latin America (after Brazil), with more than 135,000,000 inhabitants. According to the General Directorate of Health Information, Birth Certificates from Ministry of Health and Subsystem of Birth Information (SINAC), there about 2,200,000-2,500,000 live births each year. It is estimated that 12,000 to 16,000 newborns have CHD each year. Several very recent publications exist on POS [32,33,34] and according to the report of the National Institute of Statistics and Geography (INEGI) [35], CHD represents the second leading cause of death in children under 5 years of age.

The implementation of CCHD screening in Mexico has not been easy and here, we will describe three implementation efforts in this country.

#### 3.10.1. Sonora

In 2015, the Children’s Hospital of the state of Hidalgo, Mexico, with a population of close to 3,000,000, was the first to start POS. Its capital and most populous city is Hermosillo. It currently has more public hospitals that perform POS for CCHD screening than any other one in Mexico. POS started the Cardiac Screening Project at three hospitals, with the aim of making it a part of newborn care programs. In 2016, three other hospitals in the state started to implement POS as part of mandatory newborn care.

On April 30, 2017, the neonatal cardiac screening was approved as a health care initiative before the Congress of the State of Sonora, but it was not until June 2018 that it began as a pilot program at the Children’s Hospital. So far, generous donations are supporting POS.

CCHD screening in participating institutions has not only served for the detection of CCHD, but also for other diseases that present with hypoxemia, such as pulmonary hypertension and sepsis. Their early detection reduces morbidity and mortality and also the duration of hospital stays. During these years of cardiac screening in the State of Sonora, 9181 apparently healthy newborn infants were screened. Of them, 22 (0.24%) had a positive test. Of these 22 infants, 11 (50%) had congenital ductal-dependent heart lesions, 8 (36%) had pulmonary hypertension, 3 (14%) had early sepsis, and 2 (9%) were true false positive tests. This means that only 0.02% of screened infants had a true false positive result. However, these two infants had skin conditions, one of whom also presented with generalized melanosis.

Sadly, the greatest challenge identified for further implementation and the main difficulty encountered have been the obstacles posed by the doctors themselves. The leader in this effort keeps moving forward and convincing health care professionals. However, the delays are real and regrettable.

Pertinent information on the findings of positive POS with no CCHD revealed that moderate to severe persistent pulmonary hypertension of newborns was the most frequent condition found. For these eight asymptomatic infants, an echocardiogram confirmed the diagnosis. The initiation of early treatment, with oxygen supplementation and sildenafil or bosentan, was effective. None of these eight infants required mechanical ventilation or inhaled nitric oxide and most improved in 3 to 5 days, without the need for inotropes or any other medication. This has a beneficial impact on costs and in short- or long-term morbidities.

Of note is that, of the four hospitals that have a universal POS program, two of them charge 600 Mexican pesos (approximately 32 US dollars) per test; one private hospital has the charge included in the ‘maternal package’; and in the Children’s Hospital, there is no charge for newborns born there, but there is charge of 200 pesos (about 11 US dollars) for newborns born at other institutions that go there and request POS.

#### 3.10.2. Guadalajara

Guadalajara is the capital city of the state of Jalisco, with a population of about 6,000,000, which is 8,000,000 for the whole state. It has, on average, around 2500 births per year.

In 2019, the Pediatric Hospital of the Mexican Institute of Social Security (IMSS), a level III and referral hospital for CHD in the region, received 128 infants with CCHD, representing 19.07% of the total number of newborns admitted to the NICU (671). The most frequent CCHDs were pulmonary valve atresia (12%), transposition of the great arteries (10%), anomalous pulmonary venous connection (8%), and aortic coarctation (7%); 13 of these infants died before surgery and the overall mortality was 37% (47 infants).

Only four (3.1%) of the 128 newborns were diagnosed early; two by a fetal echo and then POS, and two by POS after 12-24 hours.

The IMSS has many health care facilities in the country and provides care to 80 million persons or 6 out of 10 Mexicans. It is the largest health care institution of its kind in Latin America. However, unfortunately, POS has not been widely implemented for healthy newborns at the Institute’s hospitals. In the last four years, a few centers have carried it out irregularly and inconstantly due to a lack of infrastructure, particularly due to not having pulse oximeters and/or consumables (sensors), as well as a limited commitment of neonatal health care professionals.

In a personal survey we performed in Guadalajara in IMSS centers with >1500 births per year, the major problems described were scarce human resources for education, training, and performing POS regularly with adequate data collection and analysis. All surveys mentioned that there are no SpO_2_ monitors assigned to the rooming-in area. In addition, another difficulty mentioned was that most of these centers have instituted an early discharge program, within the first 12–24 h postpartum. Nevertheless, most of them are working towards future formal implementation.

The “other side of the coin” is the situation in the private sector. In one hospital, with an average of 1200 live births per year, they have implemented POS with a Nellcor® SpO_2_ monitor. In the last 12 months (2/2019-1/2020), we analyzed data for over 1000 infants. Of them, 4.6% were not tested due to early home discharge and the unavailability of personnel who could conduct POS earlier, and 2.5% were not tested for unclear reasons. Of the infants that underwent POS, six (0.6%) failed. After evaluation by pediatric cardiology with echocardiography, two were positive, with CCHD confirmed: transposition of the great arteries and atresia of the pulmonary valve. Two infants were positive due to respiratory conditions (both with persistent pulmonary hypertension) and were promptly treated. Two were true false positives, with a rate of 2‰.

In the region of Guadalajara, there is no extra charge for POS in private institutions.

#### 3.10.3. San Luis de Potosí

In the state of San Luis Potosí, there are 58,600 live births. At the Dr. Ignacio Morones Prieto Central Hospital in 2018, the number of live births was 4197. The protocol for the early detection of CCHD with POS was implemented at this hospital. However, in other institutions that receive 4000 or more newborns, POS is not carried out on a daily basis due to a lack of established protocols and, in some, due to a lack of monitors. There is difficulty in transporting infants with CCHD to specialized centers due to a lack of available beds, since, in the country, there are few hospitals where neonatal cardiovascular surgery is performed.

#### 3.10.4. Mexico City

Mexico City is the capital of Mexico, the largest city in North America, and the fifth largest city in the world, according to the United Nations. The population of Mexico City in 2019 was 21,672,000.

It is impossible to describe the existing disparities in general, and POS in particular, in this manuscript. Mexico City has a large number of public and private hospitals; 66 general hospitals, 47 specialized hospitals, more than 7000 clinics, 542 surgical rooms, 286 clinical analysis laboratories, and more than 18,000 hospital beds. The disparities are astounding. Many of the health care facilities are operated by the government and provide basic health care for the city’s poor.

Some public hospitals have started POS, irregularly. Several private centers have also started this, but at a cost to the family. On many occasions, when parents do not feel that they should be paying for the cost of POS, the screening is not done.

SIBEN has been consulted, together with Dr. Ann Granelli, by the Senate Health Commission, for a possible addition to Article 61 of the National Health Law (“New fraction II Bis”, in relation to complex congenital heart disease and screening for its early and timely detection). By the end of 2018 and mid 2019, we had had several meetings with senators and with the President of the National Health Commission. At this time, the initiative is still being pondered upon.

### 3.11. Paraguay

Paraguay is a country with about 7,000,000 inhabitants and about 110,000 live births per year. Asunción is the capital, with a population of about 700,000, making it Paraguay’s most populous city. The first POS screening program in the country for congenital heart disease and other serious hypoxemic diseases in the neonatal period started at Hospital de Clínicas, Facultad de Ciencias Médicas-Universidad Nacional de Asunción.

In 2013, a protocol started to be developed at this hospital. Then, a pilot study was conducted to perform POS in all healthy newborns before discharge or at less than 48 hours. Upon completion, licensed RNs were subsequently trained; currently, they are the ones who perform POS.

With the support of SIBEN, we started to follow POS guidelines described in the SIBEN’s Clinical Consensus [21] to implement the screening program for all births in this hospital with about 2400 annual births.

POS is performed in healthy asymptomatic infants in pre- and post-ductal territories using the SpO_2_ monitor Radical-7 with EVE software from the Masimo corporation. This is done while the infant is in the rooming-in area before hospital discharge, or by 48 h of age. With SpO_2_ ≥ 95%, there are no additional actions taken. With SpO_2_ 90–94%, a physical exam is performed and vital signs, such as the heart rate, respiratory rate, and axillary temperature, are recorded. If the infant is asymptomatic, POS is repeated within 1 hour. With symptoms, the infant is evaluated by an attending neonatologist and admitted to a neonatology unit with continuous monitoring of the heart rate and SpO_2_, and the blood pressure is measured in all extremities. According to the findings, oxygen is provided and a chest x-ray, electrocardiogram, and CBC are performed. The same steps are followed if SpO_2_ remains <95% in the second POS. When necessary, cardiology is consulted and an echocardiogram is performed. In cases where the first POS reveals an SpO_2_ <90%, the infant is immediately admitted and similar steps to those listed above are followed with urgency.

The hospital is a referral center and most of the congenital heart diseases have an intrauterine diagnosis. There are only three hospitals that are able to solve these problems in the country. In this hospital, between inborn and outborn infants, there are approximately 25 cases per year. This is the only hospital in the nation that is performing screening.

With POS, one infant with pulmonary stenosis and several others with mild to moderate hypoxemic transient conditions were identified. 

There are still significant challenges to overcome, mainly related to resource limitations. Due to this, about 7–10% of children are missed annually and not screened. As a summary, the following are the most substantial limitations:RN staffing on weekends: There is insufficient staff in relation to clinical demands and a baby or two may be missed;Insufficient equipment: The MDs, on occasion, take the SpO_2_ monitor to the NICU to monitor a sick newborn infant;Masimo^®^ sensors last for about one month and sometimes there is unavailability for a couple of days.

### 3.12. Perú

In Perú, the population is 33,300,000 and 560,000 children are born on average every year, in 25 different regions of the country. The National Maternal Perinatal Institute (INMP) is in Lima, the capital city, which has 10,555,000 inhabitants. The institution was created 179 years ago and is a national reference center specialized in maternal perinatal medicine. It serves a population with identified risk factors, most of whom belong to low socio-economic groups from the poverty rings of Lima and different regions of the country. The number of annual deliveries at INMP is 17,000–19,000. In 2018, the neonatal mortality rate was 7‰ live births; however, in 2019, in the referral hospitals for high-risk pregnancies, the neonatal mortality rate was 15.3‰ live births. The first cause of mortality was congenital malformations, of which 39% were cardiac malformations.

The NICU has, on average, 35 neonates per day, and is usually on high demand. The intermediate unit has 90 beds. Additionally, there are rooms in which the healthy baby is together with the mother. The stay for vaginal deliveries is about 24–36 h and those born by caesarean section stay 72 h. The average number of babies per day is 150. On average, each year, there are four acute deaths in this area, without an exact cause, since no autopsies are performed.

Towards the end of 2015, SIBEN supported initial efforts regarding the future implementation of POS at The National Maternal Perinatal Institute by Dr. Davila, the newly appointed Director of Neonatal Medicine. During 2016, a resident was stimulated to perform a pilot observation in the rooming-in area as part of his thesis on the detection of CCHD by pulsioximetry. For this, he used one of the SpO_2_ monitors from the intermediate care unit, whenever possible. During the four months in which the study was conducted, he was able to evaluate 500 newborns and found no positive case. In 2017, progress was made with the help of SIBEN in education and training, but it was not possible to establish POS as a universal program due to the insufficient number of monitors. In 2018, SIBEN donated one pulse oximeter with Signal Extraction Technology (SET^®^), and the Hospital Direction assigned it to be exclusively used for POS. Having the equipment, the hospital director authorized the hiring of staff to perform POS. In November 2018, the INMP began POS for CCHD and hypoxemic conditions, with one nurse dedicating 150 h a month to the program, based on SIBEN’s Clinical Consensus [21]. Progress was rapidly made, and 584 newborns were screened in the last two months of 2018. With full implementation, 12,656 infants were screened in 2019. When the diagnosis had been previously made by a fetal ultrasound, the infant was rapidly admitted to the NICU after birth. With POS in healthy-appearing neonates, 23 positive tests have been found, with 1% of true false positives. In seven infants, the diagnosis was mild pulmonary hypertension, which resolved with immediate admission and intensive care treatment. Echocardiography was performed in 18 of the infants, and three had CHD.

Another improvement made at INMP is that, as of 2019, echocardiography is available any time that it is required. Unfortunately, the IT department does not collect all infants or provide real time information, which is truly necessary with the large number of infants cared for at the INMP. About 70% of the screened infants are in a database, but the rest are filed manually. A new format was made during 2019, and for 2020, we expect that this will be fully operative. At INMP, there is still a need to improve the CCHD registry system with a complete digital database for POS. Additionally, to ensure that human resources are always available to reach a 100% coverage of healthy newborns, another pulse oximeter fully assigned to the area is necessary to simplify performing POS on days that the census is high.

At the national level, Perú had a Neonatal Screening Law (Number 29,885) issued in July 2012, which includes universal screening for the timely detection, treatment, and monitoring of congenital hypothyroidism, congenital adrenal hyperplasia, phenylketonuria, cystic fibrosis, congenital cataracts, and hearing screens. Unfortunately, the current coverage is only 35%.

POS is starting at some institutions of the Ministry of Health in northern Trujillo, Belen and Regional Hospital, Arequipa, Cusco, Tacna, Rebagliatti Hospital, and others, but not yet with an adequate and permanent coverage, and not all of them with appropriate monitors.

In relation to POS, in July 2019, with the encouragement of INMP and SIBEN’s support, the National Technical Standard for POS was issued. However, the hope is that POS will be incorporated into the Screening Law and become mandatory for all places.

### 3.13. República Dominicana (Dominican Republic)

The República Dominicana has about 11,000,000 inhabitants and 132,500 annual live births (General Directorate of Epidemiology, Ministry of Public Health). The neonatal mortality rate of 23‰ live births, until recently, was among the highest in Latin America. In 2019, several actions were implemented to improve the care of critically ill babies. These efforts were coordinated by the National Health Service (Sistema Nacional de Salud, SNS), in agreement with the Ibero-American Society of Neonatology (SIBEN), to provide evaluations and recommendations to improve neonatal care. The main action performed by SIBEN was to develop and implement an educational program provided in frequent visits to different areas of the country. The program was for neonatologists, neonatal nurses, and residents in neonatology and cannot be described in detail here. In addition, recommendations were made to improve infrastructure and staffing for all delivery services and neonatal units. In 2019, there was a remarkable improvement in neonatal survival, with a reduction of mortality of 30% (data from the National Epidemiological Surveillance System, SINAVE). We found that CCHD represents the fourth leading cause of neonatal mortality in this country.

Santo Domingo is the capital city, with a population over 1,000,000, and the number of residents of the large metropolitan area of Great Santo Domingo is about 4,000,000, representing close to 40% of the total population of the country.

In the last 24 months (2018–2019) at Dr. Hugo Mendoza Pediatric Hospital in Santo Domingo, 102 infants with CHD were evaluated; 72 (70.5%) were referred for corrective surgery, and 30 (29.6%) had CCHD, of whom 46% were apparently healthy newborns at the time of birth.

Given the high prevalence of CCHD and as a result of the actions that are being carried out to improve neonatal survival, towards the end of 2019, we started education for the early detection of CCHD with POS, utilizing SIBEN’s Clinical Consensus [21]. We set out to raise awareness of the issue among neonatologists, nurses, and neonatal residents, covering different hospitals with maternity and newborn care and a large percentage of deliveries in the nation.

Throughout the process, we have identified the following:The large majority of Dominican newborns are cared for in hospitals of the public sector;There exists a significant disparity of neonatal care among public hospitals;There are a small number of private centers that care for newborn infants, and some of them refer ill newborns to public hospitals;A significant number of neonatologists were not cognizant about the POS technique and interpretation of results;All neonatal and rooming-in nurses were unaware of POS;Those few who knew about POS did not know about its importance for the early detection of other conditions that present with hypoxemia other than CCHD;There were not enough saturation monitors and sensors and the ones available did not meet the recommended standards for POS.

After these initial efforts, it became obvious that extensive work is required to ensure that an adequate number of trained personnel are available to carry out and interpret the test and for accurate data recording. SIBEN continues to work on this. More staff are being trained in hospitals in Santo Domingo and distant hospitals and five monitors and sensors have been donated for POS. Others have been requested by the authorities. It is hoped that preliminary POS can start at some centers by the middle of 2020 or soon thereafter. Expectantly, by 2021, many newborns with CCHD and other neonatal hypoxemic conditions will be identified early and treated appropriately.

### 3.14. Summary Table 

In Table 2, we provide a summary of the information presented here in relation to POS implementation efforts in some Latin American countries guided and/or coordinated by SIBEN.

## 4. Discussion

Latin America and the Caribbean is a large and diverse geographical region with a population of over 627,000,000 and more than 11,000,000 births per year, with significant intercountry variability (from 50,000 to 3,000,000). The neonatal mortality rate is, on average, 12‰ live births, with disparity between countries, from <3‰ to >20‰. Therefore, the needs in each country and in different regions within a country are not uniform. 

In this report, we have summarized the initiatives of the last 3–4 years for the early detection of CCHD by the implementation of POS in Latin American countries with the collaboration of SIBEN. Additionally, we have described the efforts of many colleagues in diverse countries, as well as the various and capricious unique challenges and infrastructural issues they face in their countries. As shown, SIBEN’s educational activities, recommendations, and Clinical Consensus have led to remarkable achievements. However, as shown here, there are many difficulties for implementation, some of them being similar and others significantly different among countries in this region. SIBEN is committed to continuing to collaborate to try to overcome implementational challenges to POS so that more babies in Latin America can be detected early and adequately treated in an opportune manner. 

Implementation efforts are still limited and sustainability can be a great challenge in Latin America if the necessary funds are not allocated appropriately.

The early identification of critical congenital heart disease with POS is a fundamental public health issue, which would positively impact morbidity and mortality, as well as the hospital duration. There is sufficient evidence about the cost effectiveness of POS, which is a simple, effective, and high-specificity test if done with adequate SPO_2_ monitors by trained personnel.

CHD represents the third or fourth most common cause of neonatal mortality. Therefore, increasing the access to care, and establishing quality cardiac surgical programs, could lead to a substantial drop in neonatal mortality. It is likely that in the future, better care will be provided for patients with CCHD in Latin American countries. At SIBEN, we have focused on early diagnosis and referral as a priority, and collaborating on the implementation of POS for the early detection of CCHD and other neonatal hypoxemic conditions. With these steps, significant improvements are being achieved in several areas. However, the challenges for implementation have been numerous and diverse, with some common factors identified. These factors include the scarcity of suitable pulse oximetry monitors and sensors and a deficit of personnel assigned to perform POS universally every day of the year.

Even though major improvements in health infrastructure are necessary, it seems to us that the proper allocation of resources, adequate staffing, education, changes in the attitudes of many MDs and local neonatal and pediatric societies, and, of course, political will, are more important. Although the majority of newborns are healthy, neonates are sometimes a neglected population. Many of them are not examined in detail and few controls are performed. Adequate care is not guaranteed, but this would have a great impact on their future life. Those who participate in health care administration and those assigned to deliver care to normal newborns should keep this in mind. The prevention and timely detection of serious conditions increase survival, decrease morbidity, and are associated with less health care costs in the long run. Furthermore, the less supervision there is, the more justified it is to perform POS in healthy-appearing newborns before 24 h of age, even before 12 hours, but always after the first 4 h of life, to ensure that the transition period has been finalized.

It is essential to establish stronger foundations for a better structured and more equitable delivery of neonatal care in this large and diverse geographical region. This will lead to an increased demand for improvements in the different healthcare systems that exist in most countries in Latin America and to the successful and universal implementation of POS. Implementing this low-cost method is essential for achieving the goals of reducing the mortality associated with CCHD and should ultimately reduce infant mortality. At least four pillars are clearly needed to increase universal POS. One is education, as can be seen in the findings presented in this manuscript. Another one is leadership at the local level. The third one is the acquisition of adequate SPO_2_ monitors to be used only for POS in order to guarantee that they are not taken away to provide continuous monitoring for ill infants in the NICU. The fourth one is the organization of delivery of care in order to ensure permanent staffing to perform POS every day.

Finally, the data presented here shows that POS in Latin America has also been very useful for the early detection of infants with serious hypoxemic conditions other than CCHD, like persistent pulmonary hypertension sepsis, pneumonia, and other conditions. The reason for trying to implement POS in this region, where some areas have very limited options for the care and therapy of neonates with positive screening results/diagnoses of CCHD, is twofold: a) Early detection of hypoxemic conditions other than CCHD, which leads to improvements in the overall quality of newborn care, and b) an important aim in developing countries is to advance and develop measures of care to provide the infants born there with the care that they need and require. To make progress, it is important to be able to show authorities data with actual needs. This, in turn, will increase and improve options of care and therapy for infants with CCHD.

With the findings reported here, SIBEN feels very encouraged, whilst understanding that there is still a significant need for improvement. We will continue to provide voluntary support and collaboration to contribute to improving the quality of neonatal care in Latin America.

## Figures and Tables

**Table 1 IJNS-06-00021-t001:** Results of pulse oximetry screening (POS) at Hospital Niño Jesús in Barranquilla, Colombia.

Number of Newborn Infants Screened	9241
Positive tests	38
Hypoxemic conditions and persistent pulmonary hypertension	8
Atrial septal defect	1
Atrial septal defect + pulmonary stenosis	1
Ventricular septal defect + ductus arteriosus	1
Ventricular septal defect + persistent pulmonary hypertension	1
Tetralogy of Fallot	1
Total anomalous venous return	2
Single ventricle, right	1
Single ventricle, left	2
Hypertrophy of interventricular septum	1
Tricuspid atresia + hypoplastic right ventricle	1
True false positives	18

**Table 2 IJNS-06-00021-t002:** POS implementation efforts in some Latin American countries: Summary of results.

Country and Region	Births per Year	POS	Challenges	Others Comments	Law
**Argentina*#**	685,394	Variable	Urgent need for further expansion of POS in the nation	Inter-center disparity in neonatal outcomes Approximately 7000 children are born every year with CHD	National Congenital Heart Disease Program was created to coordinate referral, transfer, treatment, and follow-up of children without insurance coverage with CHD
Buenos Aires City	34,640	Intermittent, variability	Lack of equipment (pulse oximeters) and the low availability of nursing h in rooming-in areas of the maternity hospitals	Less compliance in maternities of the private sector	Projects to incorporate CCHD in the mandatory list in the Neonatal Screening National Law # 26,279
San Luis	4000	YES	The high cost of the non-reusable sensors. In addition, early discharge of both vaginal and caesarean section births	Full implementation has been achieved for the early detection of hypoxemic conditions, not only CCHD, for 6 months. Four cases (Not CCHD) detected/1400	NO
Rosario	22,000	YES, partial	Resistance, no acceptance of the added clinical value of POS, and lack of support for the program. This is worse during weekends and holidays	Since mid 2019, POS is being performed in all public hospitals, but in only one private institution. Some private hospitals that do not perform POS refer all healthy infants to pediatric cardiology, where an evaluation is done to rule out CCHD (with charges)	NO
**Bolivia*#**	243,000	NO	Lack of specialists, and costs for the implementation of programs that can solve complex problems<break/>CHD affects 9‰ live births and 25% need surgery during the first year of life	Some centers have algorithms for the early detection of CCHD. The lack of personnel and equipment, together with the non-resolution of the problems detected, have caused this to fail	NO. Bolivia does not have any program designed by the Ministry of Health for the detection and treatment of CCHD or POS
**Chile**	250,000	Intermittent, variability	The main difficulties have been the lack of equipment and supplies and the shortage of available h of trained professionals (nurses) for POS	CHD affects 9‰ live births and 25% need surgery during the first year of life	NO. Chilean health reform (2003-5): prenatal screening and a care network centralizing the surgical resolution
**Colombia***	650,000	YES, in some cities. There are only six centers performing POS in the nation	Although early detection is cost effective, treatment and follow-up are still very expensive for the economic reality of Colombia’s health system	Barranquilla started the first POS program in the country. Over 9240 screened, 20 true positive cases, 12 had various types of CCHD (see Table 1), and 8 had other hypoxemic conditions	Mandatory POS for complex congenital heart disease in the immediate neonatal period in Resolution 3280 of 2018
**Costa Rica***	65,000	YES	The biggest challenges have been related to the decision of who should perform POS	In 33,804 births, 16 infants with CCHD were detected early. A similar number was found to have other hypoxemic conditions early, mainly PPHN and sepsis.	POS established in all public and private health centers in the country since 2016, performed between 12 and 24 h of age for all healthy newborns
**Cuba*#**	117,000	Intermittent in few neonatal centers	Limited number of sensors available, monitors frequently used to care for critically ill neonates in NICU	For CCHD diagnosis, most newborns in Cuba rely on findings of universal prenatal ultrasounds and physical exams	In Cuba, abortion has been legalized. Fetal ECHO is performed in over 99% of pregnant women; when a fetus is diagnosed with serious CHD, pregnancy is terminated in many cases
**El Salvador***	91,000	NO	Technical-financial support for implementation Resistance, lack of interest, or no positive attitudinal behavior for accepting and making changes	Hospital Nacional de la Mujer and Social Security Institute plan to start universal POS by the last trimester of 2020.	NO
**Guatemala*#**	486,497	NO	Low interest of the health system and of the professionals who care for newborns	Only one public health hospital and one private hospital established a cardiac screening program, following SIBEN’s Clinical Consensus	NO
**Honduras*#**	150,000	Partial and irregular in one neonatal center	Technical-financial support for implementation Insufficient staff and equipment Human factors	Hospital Escuela (70 births/day) starts irregularly in 2017 with SIBEN support: Seven CCHD detected in 1221 NB. Reports a decrease in the average age at which heart patients receive interventions: In 2017, 13% of cases were newborn infants; in 2018, this increased to 18%; and in 2019, this increased to 24%	NO
**Mexico***	2,500,000		Wide disparities exist in the delivery of health care in general and about POS in particular	135,000,000 inhabitants	SIBEN has consulted with Senate Health Commission for the possible addition of POS to Article 61 of the National Health Law. Passed the Senate Health Commission. Now in House of Representatives
Mexico city		Intermittent, irregular, and sporadic.	City with the largest population in the Americas	Several private centers have started POS, but at a cost to the family	
Sonora		YES	Four hospitals have a universal POS program: in two of them, there is a charge per test	One of the states where more public hospitals perform POS: 9181 infants have been screened, 22 tested + (11 CCHD, 8 PPHN and 2 sepsis)	
Guadalajara		NO, irregular and inconstantly	Lack of infrastructure, pulse oximeters and/or consumables (sensors). Limited commitment of neonatal health care professionals	There is no extra charge for POS in private institutions	
San Luis de Potosí		NO, only one center	Lack of established protocols and monitors.	Few hospitals where neonatal cardiovascular surgery is performed	
**Paraguay***	110,000	In only one neonatal center	RN staffing on weekends Insufficient equipment: MDs, on occasion, take the SpO_2_ monitor to the NICU	Hospital de Clínicas, Facultad de Ciencias Médicas-Universidad Nacional de Asunción starts POS with SIBEN’s support	NO
**Perú*#**	560,000	YES, in INMP Lima. Variable in other centers and regions	Lack of adequate and permanent coverage and not all of them had an appropriate monitor. POS is starting at some institutions of the Ministry of Health	National Technical Standard for POS was issued (INMP and SIBEN’s support)	NO, Neonatal Screening Law (Number 29,885). POS not included
**República Dominicana*#**	200,000	NO	All neonatal nurses were unaware about POS. There were not enough saturation monitors and sensors and the ones available did not meet the standards.	In 2019, started education for early detection of CCHD with POS, utilizing SIBEN’s clinical consensus.	NO, CCHD represents the fourth leading cause of neonatal mortality in this country.

* SIBEN Educational Programs for MDs and RNs and Clinical Consensus on POS; # SIBEN donation of SpO_2_ monitors and sensors.

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
