# Peer review of "CCHD Screening Implementation Efforts in Latin American Countries by the Ibero American Society of Neonatology (SIBEN)"

_2409-515X, 2020, doi:10.3390/ijns6010021_

Round 1

Reviewer 1 Report

The manuscripts reports on the efforts of implementing POS in several countries in Latin America. The data presented is interesting to all involved in neonatal care, screening and health care politics.
The manuscript is clearly structured and well written.
It fits well in the scope of the journal.
Maybe the authors could comment on the reasons of trying to implement POS even in settings where there are very limited options of care and therapy for neonates with positive screening results / diagnosis of cCHD.
Additionally, conflicts of interest should be specified.

Author Response

We thank very much Reviewer 1 and appreciate her/his comments.

1. English language and style are fine/minor spell check required: We repeated a "spell check".

2. Maybe the authors could comment on the reasons of trying to implement POS even in settings where there are very limited options of care and therapy for neonates with positive screening results / diagnosis of cCHD.

The reason for trying to implement this where there are sometimes limited options of care and therapy for neonates with positive screening results / diagnosis of cCHD is two fold: a) Early detection of hypoxemic conditions other than cCHD, which leads to improve overall quality of newborn care; b) An important aim in developing countries is to advance and develop measures of care to provide the infants born there the care they need and require. To make progress it is important to be able to show authorities data with actual needs. This, in turn, will increase and improve options of care and therapy for infants with CHD.  

3. Additionally, conflicts of interest should be specified.

Added: Line 886. Conflict of interest: Augusto Sola works part time, as VP for Medical Affairs, education and research in neonatology, with an annual salary at Masimo Corporation (Irvine, Ca). None of the other co-authors has a conflict of interest.

Reviewer 2 Report

I want to congratulate the authors on this huge and important undertaking to bring sate of the art pulse oximetry screening to Latin America and to present the successes and challenges in 13 different countries , it is an amazingly large undertaking. The style is frequently editorial rather than scientific and example the health disparities are not just described but noted to be dreadful. It is well written.

 There is a very important point that I think needs to be corrected. The use of the terms CHD and CCHD are not done very well beginning with line 69 they are frequently used interchangably and there are many of the statistical reports that are unlikley to be accurate based on this confusion in terminology. For insistance in Chile line 301 you report CHD 9% of births. This would be 10 times the expected number of CCHD you also go on to report that 25 percent of them go to surgery, this would be over 5,625 surgeries if your numbers were true , this would not be possible in any country. Please review the definitions of CCHD and CHD place them in the article and use them correctly.

 You make no comments on the effects of altitude for screening I know some areas of Latin America are at very high altitude and it would be of interest to know if that limits the effectiveness.

 And finally a comment on practice:  if you are trying to screen large numbers in a low resource environment why not use reusable equipment and follow the Tennessee Protocol which starts with a single pulse oximeter reading on a foot, if the value is over 96 then there is no need to do further testing, 96 and below would require a second test on the right hand but we found that the vast majority of babies only required a single reading this would save money and time. See attached file

Author Response

We are very thankful of the encouraging and stimulating initial comments of Reviewer 2. Our sincere heart felt gratitude!

1. The use of the terms CHD and CCHD are not done very well beginning with line 69: We reviewed and corrected.

2. in Chile line 301 you report CHD 9% of births. Thank you, and our apologies if there was an error in using % instead of ‰. It reads 9‰ live births. 

3. Effects of altitude for screening. All regions in this manuscript are at sea level or at less than 250 meters above sea level, where alveolar PO2 is not significantly less. We added 2 lines (79-80) in the manuscript.

(Commentary for Reviewer 2: We are exploring this issue of early POS, after transition, between 6-18 hours of age, in two regions: a) about 2,200 meters above sea level and b) about 4,000 meters above sea level. So far, very preliminary findings, coincide with reports in the literature and SpO2 is not less, likely due to the high % of Fetal Hb) 

4. Reusable sensors have been or are being implemented. They do not last that long and, where necessary SIBEN donates them.

5. Tennessee Protocol. Again, thank you for another valuable comment. In brief: we were aware of previous publication in J. Perinatology and the one you have sent us of 2017. As the great majority of publications describe pre- and post-ductal and Ewer describes a difference > 2%, the SIBEN's consensus group on POS adopted this one after lengthy discussions. Some were concerned in adopting the Tennessee protocol and then being criticized for "going against" what has been published in reviews and meta-analysis with hundreds of thousands of babies. A few commented on a rare baby that may have cono-truncal abnormality and SpO2 97% in the foot and 94% or less in the right hand. The issue is planned to be discussed again at our next Annual Meeting coming up in November. Thank you very much again.